# Microstructural and Optical Properties of MgAl_2_O_4_ Spinel: Effects of Mechanical Activation, Y_2_O_3_ and Graphene Additions

**DOI:** 10.3390/ma14247674

**Published:** 2021-12-13

**Authors:** Nina Obradovic, William G. Fahrenholtz, Cole Corlett, Suzana Filipovic, Marko Nikolic, Bojan A. Marinkovic, Simone Failla, Diletta Sciti, Daniele Di Rosa, Elisa Sani

**Affiliations:** 1Institute of Technical Sciences, Serbian Academy of Sciences and Arts, 11000 Belgrade, Serbia; suzana.filipovic@itn.sanu.ac.rs; 2Materials Science and Engineering, Missouri University of Science and Technology, Rolla, MO 65409, USA; billf@mst.edu (W.G.F.); colecorlett10@gmail.com (C.C.); 3Institute of Physics Belgrade, 11000 Belgrade, Serbia; nikolic@ipb.ac.rs; 4Departamento de Engenharia Química e de Materiais, Pontifícia Universidade Católica do Rio de Janeiro, Rio de Janeiro 22250-000, Brazil; bojan@puc-rio.br; 5Institute of Science and Technology for Ceramics (ISTEC), 48018 Faenza, Italy; simone.failla@istec.cnr.it (S.F.); diletta.sciti@istec.cnr.it (D.S.); 6National Institute of Optics, National Research Council (CNR-INO), 50125 Firenze, Italy; daniele.dirosa@ino.it (D.D.R.); elisa.sani@ino.it (E.S.)

**Keywords:** mechanical activation, dopants, sintering, optical properties

## Abstract

Magnesium aluminate and other alumina-based spinels attract attention due to their high hardness, high mechanical strength, and low dielectric constant. MgAl_2_O_4_ was produced by a solid-state reaction between MgO and α-Al_2_O_3_ powders. Mechanical activation for 30 min in a planetary ball mill was used to increase the reactivity of powders. Yttrium oxide and graphene were added to prevent abnormal grain growth during sintering. Samples were sintered by hot pressing under vacuum at 1450 °C. Phase composition and microstructure of sintered specimens were characterized by X-ray powder diffraction and scanning electron microscopy. Rietveld analysis revealed 100% pure spinel phase in all sintered specimens, and a decrease in crystallite size with the addition of yttria or graphene. Density measurements indicated that the mechanically activated specimen reached 99.6% relative density. Furthermore, the highest solar absorbance and highest spectral selectivity as a function of temperature were detected for the mechanically activated specimen with graphene addition. Mechanical activation is an efficient method to improve densification of MgAl_2_O_4_ prepared from mixed oxide powders, while additives improve microstructure and optical properties.

## 1. Introduction

Magnesium aluminate spinel, MgAl_2_O_4_, the only compound in the MgO-Al_2_O_3_ binary system at ambient pressure, is a ceramic of great importance in modern technologies due to its high hardness, high melting point and low dielectric constant [1]. It is attractive also due to its corrosion resistance, mechanical properties, and low cost [2,3]. When fully dense, pure MgAl_2_O_4_ can be transparent in visible light. The properties of ceramics strongly depend on the composition, nature of powders, impurities or additives, and fabrication methods [4]. Many different routes have been used to synthesize dense spinel-based ceramics such as hot pressing, pressureless methods, spark plasma, and microwave sintering utilizing spinel powder as the raw material, with or without sintering additives [5,6,7,8,9,10,11,12]. Some authors have reported on direct solid-state reactions of oxides, wet chemical precipitation, and mechanical activation [13,14,15,16,17,18,19]. The choice of synthesis method is based on the desired particle size and purity of the spinel powder. Additionally, preparation conditions have a great impact on the final microstructure and properties of ceramic materials [20].

Hot pressing (HP) has been recognized as an advanced method for ceramic fabrication. HP is a fast and efficient sintering technique that uses applied pressure and low-strain-rate powder metallurgy process for forming of a powder or powder compact at a temperature high enough to induce sintering and creep processes [21]. HP can use high heating rates (50 °C·min^−1^ or higher) and short non-isothermal sintering times (30 min or less), and it is considered an excellent method for preparing nanostructured ceramics and nanocomposites [21,22]. Compared to conventional sintering processes, HP can significantly shorten the sintering time to few minutes and can promote full densification of materials which are difficult to sinter using conventional sintering techniques. The key issue in all sintering approaches is controlling the grain growth/densification behavior. One of the possible ways to prevent grain growth during sintering is addition of very small quantities of various compounds (Y_2_O_3_, LiF, NaF, AlN, etc.). Another alternative, prior to sintering, is mechanical activation (MA), which is a high-energy ball milling process that induces physicochemical changes in spinels [16,17,18,23,24,25,26,27]. MA produces defects in materials which increase the chemical activity and accelerate mechanisms of sintering. Increasing the speed of the process decreases the sintering time and temperature [28,29]. Furthermore, mechanical activation can also affect the final physical properties of sintered bodies [30]. Such milling processes are attractive methods, because they enable the formation of submicron and/or nanostructured materials with desirable properties [31].

In the present work, we report the influence of MA, in combination with additives, on densification and the final properties of MgAl_2_O_4_ sintered bodies. We focused on studying doped spinel (concentration of the Y_2_O_3_ 0.1 wt.%, and graphene 1 wt.%) to improve their optical properties. X-ray powder diffraction (XRPD), scanning electron microscopy (SEM), and UV–visible spectroscopy were used for characterization of the as-prepared samples.

## 2. Experimental Methods

A mixture of high-purity MgO and α-Al_2_O_3_ starting powders (all 99.9% purity Sigma–Aldrich, p.a., St. Louis, MO, USA) was used in these experiments. The starting MgO and α-Al_2_O_3_ powders were added in a one-to-one molar ratio to produce stoichiometric MgAl_2_O_4_. The powders were mixed by ball milling for 1 min to homogenize them without significant particle size reduction. A portion of the as-obtained powders was additionally mechanically activated for 30 min in a high-energy planetary ball mill (Planetary Ball Mill Retsch PM 100, Haan, Germany) in air. One wt.% of graphene (99.9% purity Sigma–Aldrich, p.a., St. Louis, MO, USA) was added to a second powder mixture prior to activation, while another batch of powder contained 0.1 wt.% of Y_2_O_3_ (99.99% purity, Pangea International Ltd.-Shanghai, China). The first steps of homogenization and mechanical activation were performed using the same mill with Y-stabilized ZrO_2_ vials and balls. The media were 5 mm in diameter. The ball-to-powder weight ratio was 40:1, with a rotation speed of 400 rpm. Powders were sieved after milling. The powder mixtures were labelled as HP1 (ball milled plus 30 min mechanical activation), HP2 (ball milled only), HP3 (30 min activated with yttrium addition), and HP4 (30 min activated with graphene addition). Powders were hot pressed in a graphite die (Ø = 30 mm) at 1450 °C in vacuum (20 Pa), with an applied pressure of 30 MPa, heating ramp 30 °C·min^−1^, and free cooling and dwelling times were between 5–10 min (see Table 1).

The procedure used for synthesis and densification of the spinel materials in the present study is based on our previous manuscript [13] and briefly summarized in the present paper. Densities of sintered specimens were calculated by Archimedes’ principle. Sintered specimens were subjected to X-Ray powder diffraction (XRPD; X’Pert Pro, PANalytical, Almelo, Netherlands) in the Bragg–Brentano geometry using Cu-Kα radiation. Measurements were conducted on polished cross sections of the hot-pressed specimens. Phase analysis was performed by Rietveld refinement (RIQAS4, Materials Data Incorporated, Livermore, CA, USA). Lattice parameters determined using Rietveld refinement of XRD data were used to calculate the theoretical density of sintered bodies, assuming cubic crystal structure and the space group $\mathrm{Fd}\bar{3}m$ (227). Microstructure was examined by scanning electron microscopy (SEM; Raith eLine, Raith GmbH, Islandia, NY, USA). The samples were polished cross sections that were coated with a conductive Au/Pd coating before placing into the SEM.

Room-temperature hemispherical optical reflectance and transmittance spectra for quasinormal incidence angle have been acquired from polished cross sections using two instruments: a double-beam spectrophotometer (Perkin Elmer Lambda900, MA, USA) with a 150 mm-diameter Spectralon^®^-coated integration sphere for the 0.25–2.5 µm wavelength region, and a Fourier transform infrared spectrometer (FT-IR Bio-Rad Excalibur, CA, USA) with gold-coated integrating sphere and liquid nitrogen-cooled detector for the spectral range 2.5–15.7 µm. From experimental reflectance (R^∩^(λ)) and transmittance (T ^∩^(λ)) data, the spectral absorbance α(λ) or emittance ε(λ) can be obtained, as:α(λ) = 1 − R^∩^(λ) − T ^∩^(λ) = ε(λ)(1)

Thicknesses of samples are 6.50 mm for HP1, 6.90 mm for HP2 and HP3 and 6.95 mm for HP4.

## 3. Results and Discussion

Figure 1 shows the specimens after sintering. All specimens were disc shaped. The ones with graphene additions were black, and the rest were white or grey. They all achieved relative densities over 96%.

XRPD patterns of the sintered specimens are presented in Figure 2. Magnesium aluminate with the spinel structure is the only phase present in all samples and was identified using PDF card 01-077-1203. All peaks were well-defined, with high intensity, and are sharp, indicating high crystallinity. Rietveld analysis corroborated the presence of phase-pure spinel in all sintered specimens, with crystallographic density of 3.584 g·cm^−3^, which corresponds to its theoretical value. Lattice parameters were all about *a* = 8.077 Å. Neither mechanical activation nor additives affected lattice parameters significantly, indicating that neither carbon nor yttrium were likely substituted into the spinel lattice. Crystallite size determined by Rietveld refinement exhibited a maximum value of 429 Å for HP1, but decreased with the addition of yttrium or graphene with a minimum value of 391 Å for HP3. Incorporation of additives into spinel decreased both the lattice parameter and their crystallite size.

SEM images of sintered specimens are shown in Figure 3. All samples achieved a certain degree of translucency, indicating that the residual porosity was low [32]. Dense and homogeneous fully sintered matrix were characteristics of all samples. The nonactivated HP2 sample possesses the largest number of small, closed pores that were less than 200 nm, exhibiting a density of 3.46 g·cm^−3^ (96.5% TD). Voids larger than 1 μm were visible on the first three samples (HP1, HP2 and HP3), but were a consequence of grains being pulled out from the surface during the preparation process. A smaller number of closed spherical pores were visible on other micrographs (b, c, d), while the densest sample was the one with the yttrium addition (HP3), which reached more than 99.5% TD.

The spectral absorbance of samples is shown in Figure 4. The bands above ~11 µm wavelength were apparent in all samples, albeit with different depths. The relative differences between samples HP1-2-3 were due to compositional variations, such as the presence of Y_2_O_3_ and the contamination from milling media. The addition of graphene (sample HP4) increased the spinel spectral absorbance in the whole investigated wavelength range, and completely removed the complex spectral features shown by the other samples below 6 µm, bringing the curve to an almost flat, high-reflectance plateau. This major effect was obtained through both a reduction in the spectral reflectance in the range 0.3–6.0 µm (Figure 5a) and the zeroing of near-infrared spinel transparency window (see the transmittance curves in Figure 5b).

From the spectra, some parameters can be calculated to evaluate the potential of the materials as sunlight absorbers in solar receivers for thermodynamic solar plants. These parameters are solar absorptance *α*, total hemispherical emittance *ε* at the temperature *T*, and spectral selectivity *α*/*ε*, and are expressed by the following relationships:(2)$$\alpha =\frac{\int_{0.3\mu m}^{3\mu m}\alpha \left(\lambda \right)\cdot S\left(\lambda \right)d\lambda }{\int_{0.3\mu m}^{3\mu m}S\left(\lambda \right)d\lambda }$$
(3)$$\varepsilon =\frac{\int_{0.3\mu m}^{15.7\mu m}\varepsilon \left(\lambda \right)\cdot B\left(\lambda ,T\right)d\lambda }{\int_{0.3\mu m}^{15.7\mu m}B\left(\lambda ,T\right)d\lambda }$$
where *S(λ)* is the sunlight spectral distribution [33] and *B(λ,T)* is the blackbody spectral radiance at the temperature *T*. For a more complete evaluation, the emittance in Equation (3) has been calculated at different temperatures from 800 to 1500 K. As a methodology comment referring to these parameters, the values calculated from Equations (1)–(3) were obtained from room-temperature spectra. These values are an estimation that is widely used in the literature for a comparative evaluation among materials. As such, they are not the exact values in operative conditions, which would need spectra acquired at the considered temperature. The methodology used herein generally underestimates the values of *α* and *ε* [34,35,36].

The calculated α values are listed in Table 2. In addition, data for two reference samples of dense SiC were also added [37,38] because SiC is an advanced high-temperature solar receiver material that is currently employed in existing plants.

For an ideal solar absorber, α should be as close as possible to unity. From Table 2, the samples with the highest performance are HP4 and HP1, showing α values equal (HP1) or even better (HP4) than the best SiC. HP2 and HP3 can also be considered good solar absorbers, as their α values are comparable to those of the dense polished SiC [37].

When thermal emittance (Equation (3)) is considered, Figure 6 compares the spinel samples to the SiC pellets [39], showing that HP4 had black-body-like behaviour with high thermal emittance, which is attributed to the graphene addition. The other spinel samples had emittance values similar to each other and comparable to SiC at temperatures above ~900 K. The spectral selectivity (α/ε ratio, Figure 7) shows that, at high temperatures, HP2 had a spectral selectivity similar to the more spectrally selective SiC specimens, while HP4 was similar to the second SiC reference pellet. Therefore, HP4 appears to be the most promising material from the present work for high-temperature solar thermal receivers, and should be comparable to SiC from the viewpoint of optical parameters.

Finally, given the spectral characteristics of the reflectance and transmittance spectra (Figure 5a,b) and considering that the range of nonzero transmittance is largely located in the infrared, outside the sunlight spectral region, increasing the transmittance by reducing the ceramic thickness could be beneficial in terms of optical parameters, as it reduces ε while only negligibly decreasing α, thus increasing the α/ε ratio. The optical performance of HP1, HP2 and HP3 samples could be improved using thinner specimens, combined, if needed for structural resistance, with a low-emittance substrate (e.g., a metal).

## 4. Conclusions

The influence of MA and additives was studied regarding the synthesis of MgAl_2_O_4_ spinel and its final properties. The focus was to monitor the influence of 30 min of mechanical activation along with additions of 0.1 wt.% Y_2_O_3_ and 1.0 wt.% graphene on changes in the microstructure and optical properties. Both MA and additives are beneficial, and the main conclusions are:All specimens were nominally phase-pure spinel after HP. Dense, homogeneous and fully sintered microstructures were characteristic of all specimens. Nonactivated samples exhibited a larger number of closed spherical pores and lower relative densities (96.5% TD), while the densest sample was the one with Y_2_O_3_ addition (>99.5% TD).The samples with the highest optical performance were HP4 and HP1, showing α values equal to (HP1) or even better (HP4) than reported values for SiC. All samples could be good candidates for solar absorbers, but the sample with graphene addition appears to be the most promising sample for high-temperature solar thermal receivers. The optical parameters were comparable to SiC.

## Figures and Tables

**Figure 1 materials-14-07674-f001:**
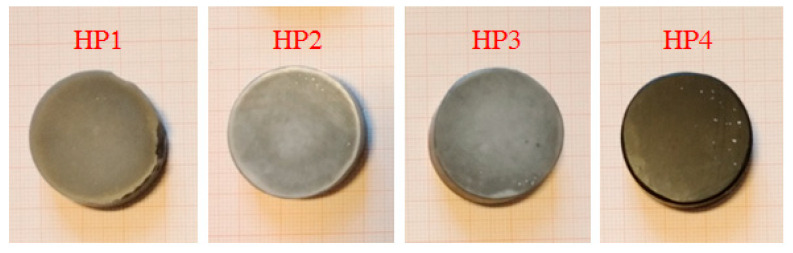
Sintered specimens: HP1—30 min-activated sample, HP2—nonactivated sample, HP3—30 min-activated sample with yttrium addition, and HP4—30 min-activated sample with graphene addition.

**Figure 2 materials-14-07674-f002:**
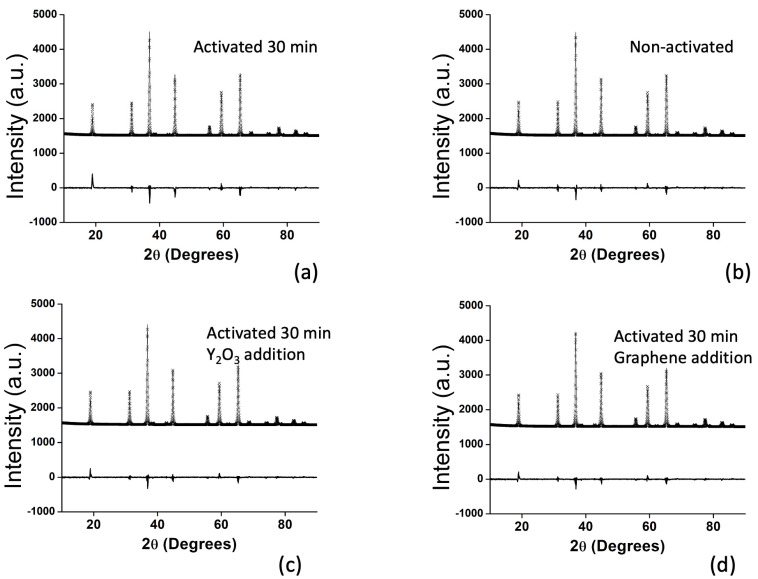
XRPD patterns and Rietveld fits of the sintered specimens: (**a**) HP1, (**b**) HP2, (**c**) HP3, and (**d**) HP4.

**Figure 3 materials-14-07674-f003:**
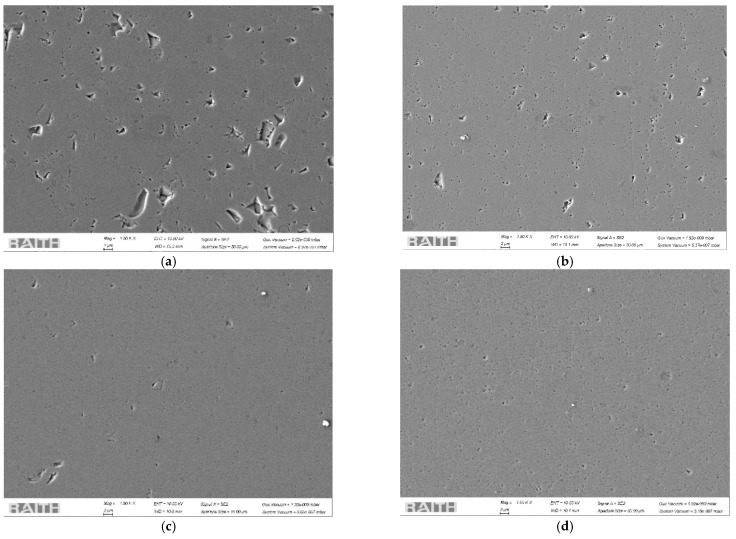
SEM images of sintered specimens: (**a**) HP1, (**b**) HP2, (**c**) HP3, and (**d**) HP4.

**Figure 4 materials-14-07674-f004:**
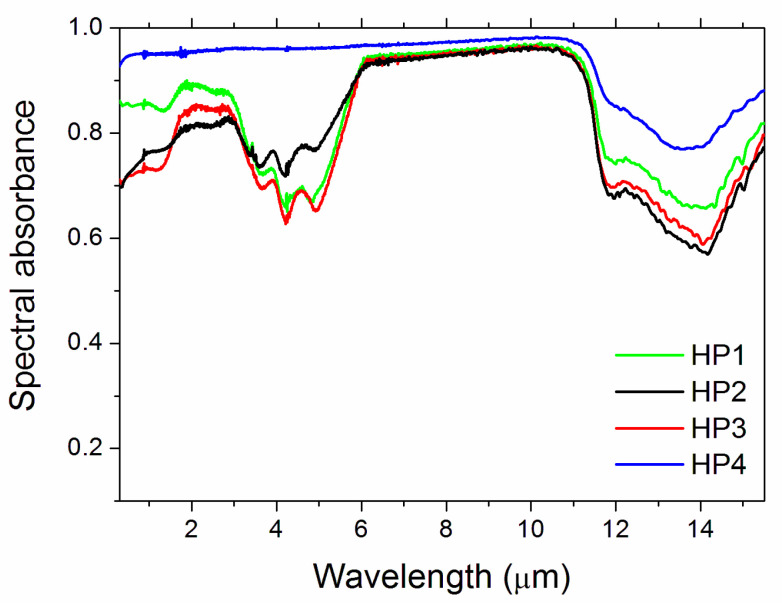
Room-temperature spectral absorbance.

**Figure 5 materials-14-07674-f005:**
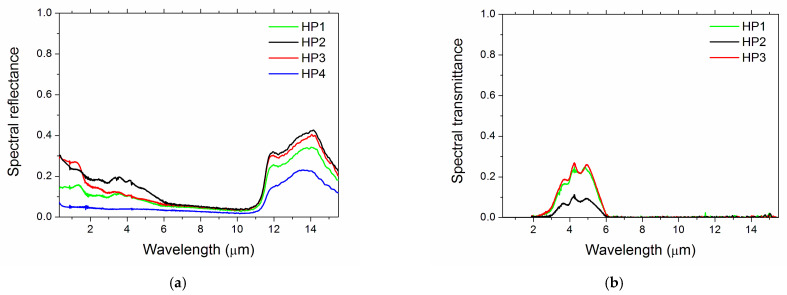
Room-temperature spectral hemispherical reflectance (**a**) and spectral hemispherical transmittance (**b**). Sample HP4 shows a null transmittance in the whole investigated range, thus it does not appear in the plot.

**Figure 6 materials-14-07674-f006:**
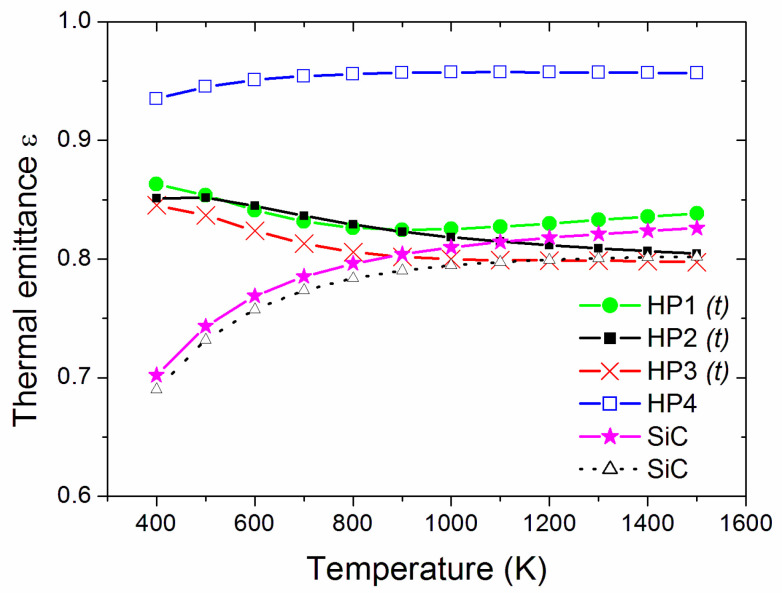
Estimated thermal emittance as a function of temperature. The label *(t)* identifies the samples showing transparency (non-null transmittance) at the considered thicknesses.

**Figure 7 materials-14-07674-f007:**
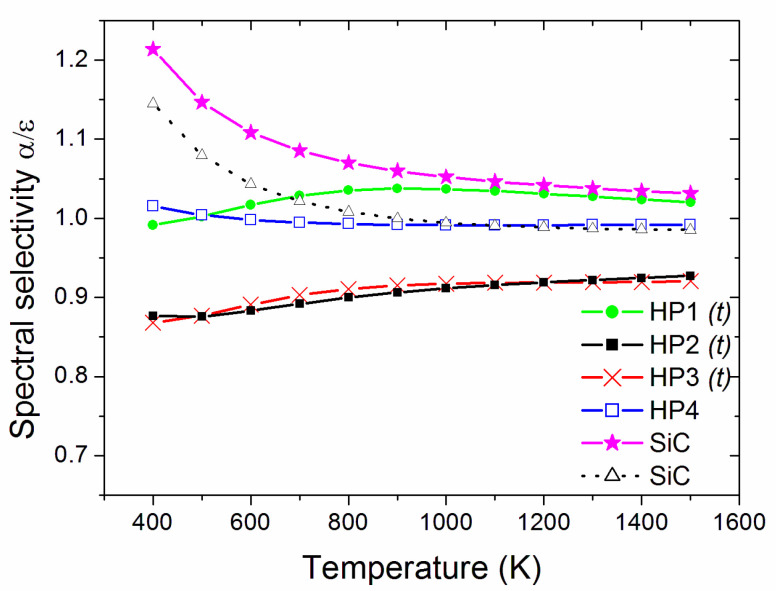
Estimated temperature-dependent spectral selectivity. The label *(t)* identifies the samples showing transparency (non-null transmittance) at the considered thicknesses.

**Table 1 materials-14-07674-t001:** Starting composition, sintering parameters, densities.

Name	Composition	Process	Temperature	Pressure	Dwell Time at T_MAX_	Height	Weight	Bulk Density
			°C	MPa	min	mm	g	g·cm^−3^
HP1	Al_2_O_3_ + MgO	Planetary	1450	30	5	7.5	19.35	3.52
HP2	Al_2_O_3_ + MgO	Ball milled only			10	8.04	18.84	3.46
HP3	Al_2_O_3_ + MgO + 0.1 wt.% Y_2_O_3_	Planetary			5	7.1	18.55	3.55
HP4	Al_2_O_3_ + MgO + 1 wt.% C	Planetary			6	7.5	19.49	3.47

**Table 2 materials-14-07674-t002:** Solar absorptance of sintered specimens, calculated from Equation (2) and of SiC pellets from [37,38].

Sample	Solar Absorptance α
HP1	0.85
HP2	0.74
HP3	0.73
HP4	0.95
Dense SiC [37]	0.78
Dense SiC [38]	0.85

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
