# Peer review of "Microstructural and Optical Properties of MgAl2O4 Spinel: Effects of Mechanical Activation, Y2O3 and Graphene Additions"

_materials, 2021, doi:10.3390/ma14247674_

Round 1

Reviewer 1 Report

The presented paper deals with the manufacturing of Y2O3 and graphene doped MgAl2O4 ceramics for application in solar recievers. The stufy is scientifically sound in the most parts and well written. After a minor revision and discussion of some points (see below) it is suitable for publication in Materials.

Title: Y3+ or Y2O3 addition; present title suggests, that elemental yttrium was used
ll 35-36: this statement is only correct for ambient conditions; at higher pressure orthorhombic Mg2Al2O5 exists as another phase
l 59: typo " altrenative" --> alternative
l 65: change to "AN attractive method"
ll 76-77: please provide some experimental details for the initial ball milling step (apparatus, ball diameter, ball to powder ratio, time, rpm)
table 1: what does "100-65-01" stand for?
ll100-101: please provide information about the measurement geometry (Bragg-Brentano reflection on intact discs, transmission on grinded material?) and the radiation used; please provide a reference to the RIQAS 4 software used for the Rietveld analyses
general question on XRD, SEM and optical characterization: Was the sample surface treated in any way (grinding, polishing) before the measurements, or were the samples used as obtained after hot pressing?
figure 2: please provide all plots with the fitted curve and the difference plot between experimental pattern and Rietveld fit
ll 136-137: The number of decimals given for the lattice parameters should be critically reflected: five digits after the decimal separator induces an accuracy of 0.00001 Angström, or 0.001 pm; the size of an oxide ion is 140 pm! Do not be confused by the "accuracy" you obtain from the Rietveld software.From my expierience, more than three digits after the separator (for Angström unit) are not reasonable. You should round yor numbers accordingly. Therefore, the variance of the lattice parameter is clearly lower than the experimental error due to geometric errors if the diffractometer as well as sample height error (if reflection measurements were done). 
figure 3: you should also provide SEM images with higher magnification which show the grain size in your samples
general question on the application as solar reciever: Apparently, graphen improves the solar absorptance significally. Nevertheless, the question of graphene stability inside the spinel samples at elevated temperature must be discussed. In air, carbon oxidizes at 600-700 °C, how will the graphene inside your samples behave? Alternatively, the solar absorber has to be used in inert gas atmosphere. These quustions should be discussed in your paper!
The formatting of references is not in the Materials formatting style.

Author Response

Dear Editor,

First, I would like to thank you and the reviewers for accepting the manuscript for possible consideration, and useful changes and suggestions. Here is the list of changes that we entered in our paper:

  • Title: Y3+ or Y2O3 addition; present title suggests, that elemental yttrium was used

The title was changed to address this concern by replacing Yttrium with Y2O3.

  • ll 35-36: this statement is only correct for ambient conditions; at higher pressure orthorhombic Mg2Al2O5 exists as another phase

The pressure during hot pressing is 32 MPa, which is not high enough to promote the phase transitions that have been reported. For example, Xu et al. (Applied Physics Letters 2020;117:04901) reported that the pressure for the cubic-to-tetragonal phase transition in spinel occurred at 25 GPa, about three orders of magnitude higher pressure than was used for densification in the present study.  Hence, we do not believe that the pressures are high enough to induce any phase changes.  To clarify this issue, the word "high" has been removed and the text altered to prevent such confusion.

  • l 59: typo " altrenative" --> alternative

The typo was fixed

  • l 65: change to "AN attractive method"

The change was made to "...are attractive methods..." to address this issue.

  • ll 76-77: please provide some experimental details for the initial ball milling step (apparatus, ball diameter, ball to powder ratio, time, rpm)

same mill was used as for additional milling, only 1 min, enough for homogenizing the powder mixture.

  • table 1: what does "100-65-01" stand for?

That was a weight ratio between Al2O3 – MgO – Y2O3, it is removed from the table. 

  • ll100-101: please provide information about the measurement geometry (Bragg-Brentano reflection on intact discs, transmission on grinded material?) and the radiation used; please provide a reference to the RIQAS 4 software used for the Rietveld analyses

The text has been revised to provide the information requested on the geometry and specimens.  However, a reference was not provided for the Riqas software.  We do not feel this is necessary since the software is a commercial product and the supplier information was already included in the text. 

  • general question on XRD, SEM and optical characterization: Was the sample surface treated in any way (grinding, polishing) before the measurements, or were the samples used as obtained after hot pressing?

The text has been revised to include the requested information.

  • figure 2: please provide all plots with the fitted curve and the difference plot between experimental pattern and Rietveld fit

All plots are presented with the fitted curve

  • ll 136-137: The number of decimals given for the lattice parameters should be critically reflected: five digits after the decimal separator induces an accuracy of 0.00001 Angström, or 0.001 pm; the size of an oxide ion is 140 pm! Do not be confused by the "accuracy" you obtain from the Rietveld software. From my expierience, more than three digits after the separator (for Angström unit) are not reasonable. You should
    round yor numbers accordingly. Therefore, the variance of the lattice parameter is clearly lower than the experimental error due to geometric errors if the diffractometer as well as sample height error (if reflection measurements were done).

The lattice parameter values have been rounded according to the reviewer suggestions and the text has been adjusted to indicate that all of the lattice parameter values were the same within measurement error.

  • figure 3: you should also provide SEM images with higher magnification which show the grain size in your samples

The reviewer has requested higher magnification images, but in this case higher magnification is not needed to reveal the grain sizes. The surfaces would have to be etched to reveal individual grains. Such a procedure was not part of the current research since the goal of the micrographs was to support the relative density measurements. None of the conclusions of the paper require precise grain size measurements. Adding additional micrographs and analysis would only add to the length of the paper and would not support the content or conclusions.  In addition, this project has concluded and the specimens are no longer available to us for analysis.

  • general question on the application as solar reciever: Apparently, graphen improves the solar absorptance significally. Nevertheless, the question of graphene stability inside the spinel samples at elevated temperature must be discussed. In air, carbon oxidizes at 600-700 °C, how will the graphene inside your samples behave? Alternatively, the solar absorber has to be used in inert gas atmosphere. These quustions
    should be discussed in your paper!

Graphene incorporated into a dense ceramic would be protected from oxidation.  Any oxygen would have to diffuse through the ceramic to get to the graphene, then any gases generated by oxidation would have to diffuse out of the ceramic to the surface.  

  • The formatting of references is not in the Materials formatting style.

References are formatted according to style of the journal.

I would like to ask you to fix Figure 5b. Letter b on the figure is over Spectral transmittance.

We hope you will find these changes satisfactory and accept our paper for publication.

Best regards,

Dr. Nina Obradovic

Reviewer 2 Report

The article is devoted to the study of the properties of Magnesium aluminate spinel obtained using the method of mechanochemical synthesis followed by thermal sintering. This line of research is of scientific interest and may be of interest to a wide range of readers and researchers involved in such experiments. In my opinion, the work can be accepted for publication after the authors answer a number of questions that arose while reading this article.

1. The authors should give a more detailed description of the choice of conditions for solid-phase synthesis, in particular, the temperature and time of annealing for crystallization.
2. Have the authors established substitutional solid solutions, or did the low concentration of dopants not lead to the formation of complex phases?
3. The change in the grain size as a result of doping indicates the ordering of the structure, but the authors in this case need to indicate how this affects the dislocation density and deformation of the crystal structure?
4. SEM images require additional explanations, whether the porosity of the obtained samples was estimated by analyzing the images obtained, or whether it was determined by the method of X-ray diffraction.
5. The appearance of the presented diffraction patterns requires quality improvement.

Author Response

1) More details on the synthesis and crystallization of the ceramics for this study are reported in our previous paper, which is listed as reference 13 in the present paper.  To avoid overlap with that paper, we have not provided all of the details in the present paper.  To address the reviewer comment, we have added a sentence to the procedure section to clarify the processing conditions and to guide readers to the previous paper that describes the choice of parameters in more detail.

2) As stated on lines 138 and 139 in the manuscript, we believe that the additives were present as second phases and did not form substitutional solid solutions.  This assertion is supported by the lattice parameters of the four different specimens, which were all the same within standard deviation.  Also, we do not believe that complex phases were formed, but that the graphene and Y2O3 remained in their original forms after densification.  The low volume fraction of these phases did not allow them to be detected by XRD.

3) We do not understand this comment.  The crystallite size that was based on Rietveld refinement of XRD data was presented in the text and changed depending on the additives and milling process.  The changes are attributed to a combination of the change in particle size due to milling and the physical effect of pinning of grain growth by the added phases (graphene or Y2O3).  The scope of this study did not involve deformation mechanisms or defect characterization, so we are not sure why the reviewer asked for characterization of deformation and dislocation density, which would require both experiments to mechanically deform the specimens and extensive TEM characterization that were not part of the research that was conducted.

4)  The amount of porosity in the specimens was calculated using bulk density measurements made using Archimedes principle that were summarized in Table I and the true density that was calculated from the XRD lattice parameter measurements.  This is described in the procedure, so no changes were made to the present version based on this comment.

5)  Figure 2, the XRD figure, has been revised and updated with better quality plots.

We hope you will find these changes satisfactory and accept our paper for publication.

Best regards,

Dr. Nina Obradovic
